# Test-Time Personalization with a Transformer for Human Pose Estimation

**Yizhuo Li**[*]
Shanghai Jiao Tong University
liyizhuo@sjtu.edu.cn

**Miao Hao**[*]
UC San Diego
mhao@ucsd.edu

**Zonglin Di**[*]
UC San Diego
zodi@ucsd.edu

**Nitesh B. Gundavarapu**
UC San Diego
nbgundav@ucsd.edu

**Xiaolong Wang**
UC San Diego
xiw012@ucsd.edu

## Abstract

We propose to personalize a 2D human pose estimator given a set of test images of a person without using any manual annotations. While there is a significant advancement in human pose estimation, it is still very challenging for a model to generalize to different unknown environments and unseen persons. Instead of using a fixed model for every test case, we adapt our pose estimator during test time to exploit person-specific information. We first train our model on diverse data with both a supervised and a self-supervised pose estimation objectives jointly. We use a Transformer model to build a transformation between the self-supervised keypoints and the supervised keypoints. During test time, we personalize and adapt our model by fine-tuning with the self-supervised objective. The pose is then improved by transforming the updated self-supervised keypoints. We experiment with multiple datasets and show significant improvements on pose estimations with our self-supervised personalization. Project page with code is available at https://liyz15.github.io/TTP/.

## 1   Introduction

Recent years have witnessed a large advancement in human pose estimation. A lot of efforts have been spent on learning a generic deep network on large-scale human pose datasets to handle diverse appearance changes [59, 64, 7, 15, 43]. Instead of learning a generic model, another line of research is to personalize and customize human pose estimation for a single subject [10]. For a specific person, we can usually have a long video (e.g., instructional videos, news videos) or multiple photos from personal devices. With these data, we can adapt the model to capture the person-specific features for improving pose estimation and handling occlusion and unusual poses. However, the cost of labeling large-scale data for just one person is high and unrealistic.

In this paper, we propose to personalize human pose estimation with unlabeled video data during test time, namely, *Test-Time Personalization*. Our setting falls in the general paradigm of Test-Time Adaptation [58, 35, 61, 69], where a generic model is first trained with diverse data, and then it is fine-tuned to adapt to a specific instance during test time without using human supervision. This allows the model to generalize to out-of-distribution data and preserves privacy when training is distributed. Specifically, Sun et al. [58] propose to generalize image classification by performing joint training with a semantic classification task and a self-supervised image rotation prediction task [18]. During inference, the shared network representation is fine-tuned on the test instance with the self-supervisory signal for adaptation. While the empirical result is encouraging, it is unclear how

---

[*]Equal contribution

35th Conference on Neural Information Processing Systems (NeurIPS 2021).

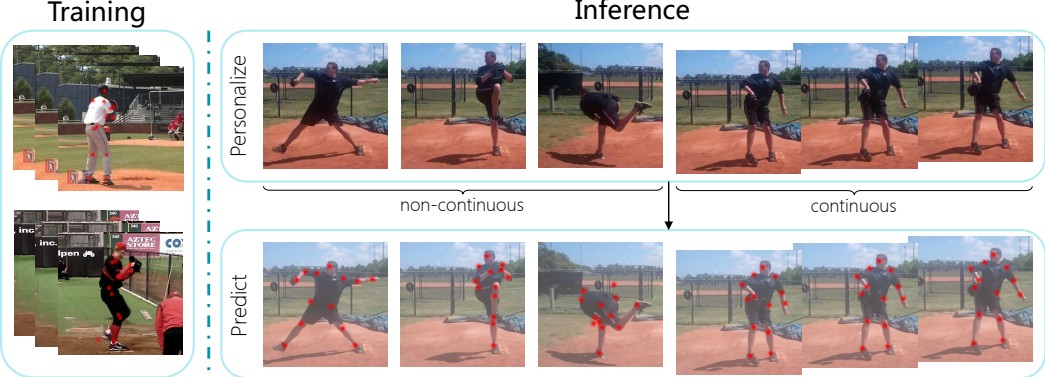

Figure 1: **Test-Time Personalization**. Our model is firstly trained on diverse data with both supervised and self-supervised keypoint estimation tasks. During test time, we personalize the model using only the self-supervised target in single person domain and then predict with the updated model. During Test-Time Personalization, no continuous data is required but only unlabeled samples belonging to the same person are needed. Our method boosts performance at test time without costly labeling or sacrificing privacy.

the rotation prediction task can help image classification, and what is the relation between two tasks besides sharing the same feature backbone.

Going beyond feature sharing with two distinct tasks, we introduce to perform joint supervised and self-supervised human keypoint estimation [26] tasks where the supervised keypoint outputs are directly transformed from the self-supervised keypoints using a Transformer [60]. In this way, when fine-tuning with the self-supervised task in test time, the supervised pose estimation can be improved by transforming from the improved self-supervised keypoints.

We adapt the self-supervised keypoint estimation task proposed by Jakab et al. [26]. The task is built on the assumption that the human usually maintains the appearance but changes poses across time in a video. Given a video frame, it trains a network to extract a tight bottleneck in the form of sparse spatial heatmaps, which only contain pose information without appearance. The training objective is to reconstruct the same frame by combining the bottleneck heatmaps and the appearance feature extracted from another frame. Note while this framework can extract keypoints to represent the human structure, they are not aligned with the semantic keypoints defined in human pose estimation. Building on this model, we add an extra keypoint estimation objective which is trained with human supervision. Instead of simply sharing features between two objectives as [58], we train a Transformer model on top of the feature backbone to extract the relation and affinity matrix between the self-supervised keypoint heatmap and the supervised keypoint heatmap. We then use the affinity matrix to transform the self-supervised keypoints as the supervised keypoint outputs. With our Transformer design, it not only increases the correlation between two tasks when training but also improves Test-Time Personalization as changing one output will directly contribute to the the output of another task.

We perform our experiments with multiple human pose estimation datasets including Human 3.6M [24], Penn Action [71], and BBC Pose [8] datasets. As shown in Figure 1, our Test-Time Personalization can perform on frames that continuously exist in a video and also with frames that are non-continuous as long as they are for the same person. We show that by using our approach for personalizing human pose estimation in test time, we achieve significant improvements over baselines in all datasets. More interestingly, the performance of our method improves with more video frames appearing online for the same person during test time.

## 2   Related Work

**Human Pose Estimation.** Human pose estimation has been extensively studied and achieved great advancements in the past few years [59, 64, 7, 15, 43, 67, 45, 21, 65, 13, 57, 75, 44, 5, 14]. For example, Toshev et al. [59] propose to regress the keypoint locations from the input images. Instead of direct location regression, Wei et al. [64] propose to apply a cascade framework for coarse to fine heatmap prediction and achieve significant improvement. Building on this line of research, Xiao et al. [65] provides a simple and good practice on heatmap-based pose estimation, which is utilized as

our baseline model. While in our experiments we utilize video data for training, our model is a single-image pose estimator and it is fundamentally different from video pose estimation models [1, 19, 62] which take multiple continuous frames as inputs. This gives our model the flexibility to perform pose estimation on static images and thus it is not directly comparable to approaches with video inputs. Our work is also related to personalization on human pose estimation from Charles et al. [10], which uses multiple temporal and continuity constraints to propagate the keypoints to generate more training data. Instead of tracking keypoints, we use a self-supervised objective to perform personalization in test time. Our method is not restricted to the continuity between close frames, and the self-supervision can be applied on any two frames far away in a video as long as they belong to the same person.

**Test-Time Adaptation.** Our personalization setting falls into the paradigm of Test-Time Adaptation which is recently proposed in [51, 50, 3, 58, 35, 61, 69, 28, 42, 20] for generalization to out-of-distribution test data. For example, Shocher et al. [51] propose a super-resolution framework which is only trained during test time with a single image via down-scaling the image to create training pairs. Wang et al. [61] introduce to use entropy of the classification probability distribution to provide fine-tuning signals when given a test image. Instead of optimizing the main task itself during test time, Sun et al. [58] propose to utilize a self-supervised rotation prediction task to help improve the visual representation during inference, which indirectly improves semantic classification. Going beyond image classification, Joo et al. [30] propose a method that proposes test time optimization for 3D human body regression. In our work for pose personalization, we try to bridge the self-supervised and supervised objectives close. We leverage a self-supervised keypoint estimation task and transform the self-supervised keypoints to supervised keypoints via a Transformer model. In this way, training with self-supervision will directly improve the supervised keypoint outputs.

**Self-supervised Keypoint Estimation.** There are a lot of recent developments on learning keypoint representations with self-supervision [55, 72, 26, 38, 32, 27, 68, 36, 40]. For example, Jakab et al. [26] propose a video frame reconstruction task which disentangles the appearance feature and keypoint structure in the bottleneck. This work is then extended for control and Reinforcement Learning [32, 36, 40], and the keypoints can be mapped to manual defined human pose via adding adversarial learning loss [27]. While the results are encouraging, most of the results are reported in relatively simple scenes and environments. In our paper, by leveraging the self-supervised task together with the supervised task, we can perform human pose personalization on images in the wild.

**Transformers.** Transformer has been widely applied in both language processing [60, 16] and computer vision tasks [63, 46, 23, 49, 56, 17, 11, 4, 73, 6, 37], specifically for pose estimation recently [66, 54, 41, 33]. For example, Li et al. [33] propose to utilize the encoder-decoder model in Transformers to perform keypoint regression, which allows for more general-purpose applications and requires less priors in architecture design. Inspired by these works, we apply Transformer to reason the relation and mapping between the supervised and self-supervised keypoints.

## 3    Method

Our method aims at generalizing better for pose estimation on a single image by personalizing with unlabeled data. The model is firstly trained with diverse data on both a supervised pose estimation task and a self-supervised keypoint estimation task, using a proposed Transformer design to model the relation between two tasks. During inference, the model conducts Test-Time Personalization which only requires the self-supervised keypoint estimation task, boosting performance without costly labeling or sacrificing privacy. The whole pipeline is shown in Figure 2.

### 3.1    Joint Training for Pose Estimation with a Transformer

Given a set of $N$ labeled images of a single person $\mathbf{I} = \{I_1, I_2 \ldots, I_N\}$, a shared encoder $\phi$ maps them into feature space $\mathbf{F} = \{F_1, F_2 \ldots, F_N\}$, which is shared by both a supervised and a self-supervised keypoint estimation tasks. We introduce both tasks and the joint framework as follows.

### 3.1.1    Self-supervised Keypoint Estimation

For the self-supervised task, we build upon the work from Jakab et al. [26] which uses an image reconstruction task to perform disentanglement of human structure and appearance, which leads to self-supervised keypoints as intermediate results. Given two images of a single person $I_s$ and $I_t$, the task aims at reconstructing $I_t$ using structural keypoint information from target $I_t$ and appearance information from source $I_s$. The appearance information $F_s^{\mathrm{app}}$ of source image $I_s$ is extracted with a

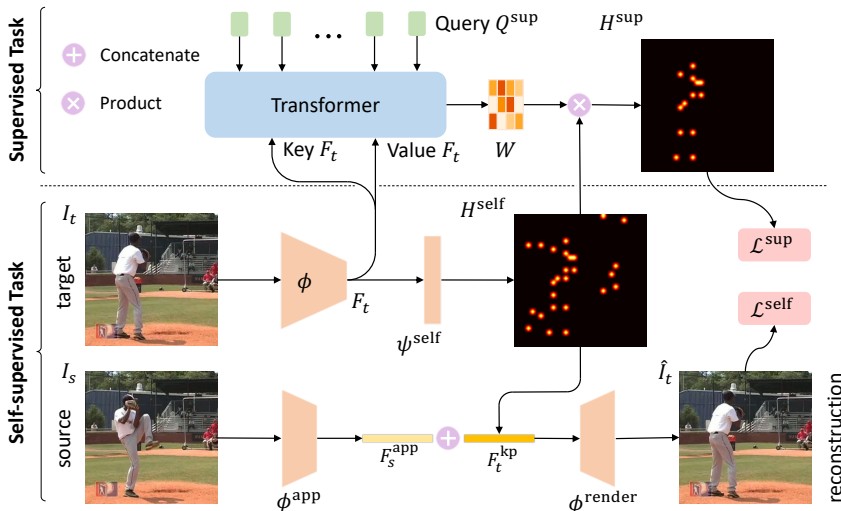

Figure 2: The proposed pipeline. 1) **Self-supervised task for personalization.** In the middle stream, the encoder $\phi$ encodes the target image into feature $F_t$. Then $F_t$ is fed into the self-supervised head $\psi^{\text{self}}$ obtaining self-supervised keypoint heatmaps $H^{\text{self}}$. Passing $H^{\text{self}}$ into a keypoint encoder (skipped in the figure) leads to keypoint feature $F_t^{\text{kp}}$. In the bottom stream, a source image is forwarded to an appearance extractor $\phi^{\text{app}}$ which leads to appearance feature $F_t^{\text{app}}$. Together, a decoder reconstructs the target image using concatenated $F_s^{\text{app}}$ and $F_t^{\text{kp}}$. 2) **Supervised task with Transformer.** On the top stream, a Transformer predicts an affinity matrix given learnable keypoint queries $Q^{\text{sup}}$ and $F_t$. The final supervised heatmaps $H^{\text{sup}}$ is given as weighted sum of $H^{\text{self}}$ using $W$.

simple extractor $\phi^{\text{app}}$ (see the bottom stream in Figure 2). The extraction of keypoints information from the target image follows three steps as below (also the see the middle stream in Figure 2).

Firstly, the target image $I_t$ is forwarded to the encoder $\phi$ to obtain shared feature $F_t$. The self-supervised head $\psi^{\text{self}}$ further encodes the shared feature $F_t$ into heatmaps $H_t^{\text{self}}$. Note the number of channels in the heatmap $H_t^{\text{self}}$ is equal to the number of self-supervised keypoints. Secondly, $H_t^{\text{self}}$ is normalized using a $\mathrm{Softmax}$ function and thus becomes condensed keypoints. In the third step, the heatmaps are replaced with fixed Gaussian distribution centered at condensed points, which serves as keypoint information $F_t^{\text{kp}}$. These three steps ensure a bottleneck of keypoint information, ensuring there is not enough capacity to encode appearance features to avoid trivial solutions.

The objective of the self-supervised task is to reconstruct the target image with a decoder using both appearance and keypoint features: $\hat{I}_t = \phi^{\text{render}}\left(F_s^{\text{app}}, F_t^{\text{kp}}\right)$. Since the bottleneck structure from the target stream limits the information to be passed in the form of keypoints, the image reconstruction enforces the disentanglement and the network has to borrow appearance information from source stream. The Perceptual loss [29] and L2 distance are utilized as the reconstruction objective,

$$\mathcal{L}^{\text{self}} = \mathrm{PerceptualLoss}\left(I_t, \hat{I}_t\right) + \left\|I_t - \hat{I}_t\right\|^2 \tag{1}$$

Instead of self-supervised tasks like image rotation prediction [18] or colorization [70], choosing an explicitly related self-supervised key-point task in joint training naturally preserves or even improves performance, and it is more beneficial to test-time personalization. Attention should be paid that our method requires only label of one single image and unlabeled samples belonging to the same person. Compared to multiple labeled samples of the same person or even more costly consecutively labeled video, acquiring such data is much more easier and efficient.

### 3.1.2 Supervised Keypoint Estimation with a Transformer

A natural and basic choice for supervised keypoint estimation is to use an unshared supervised head $\psi^{\text{sup}}$ to predict supervised keypoints based on $F_t$. However, despite the effectiveness of multi-task learning on two pose estimation tasks, their relation still stays plain on the surface. As similar tasks do not necessarily help each other even when sharing features, we propose to use a Transformer decoder to further strengthen their coupling. The Transformer decoder models the relation between

two tasks by learning an affinity matrix between the supervised and the self-supervised keypoint heatmaps.

Given the target image $I_t$, its feature $F_t$ and self-supervised heatmap $H_t^{\text{self}} \in \mathbb{R}^{h \times w \times k^{\text{self}}}$ are extracted using encoder $\phi$ and self-supervised head $\psi^{\text{self}}$ respectively, where $h, w, k^{\text{self}}$ are the height, width and number of keypoints of the heatmap. The Transformer module learns the affinity matrix based on learnable supervised keypoint queries $Q^{\text{sup}} \in \mathbb{R}^{k^{\text{sup}} \times c}$ and context feature $F_t$.

A standard transformer decoder layer consists of a multi-head attention layer and a feed-forward network. The spatial feature $F_t$ is flattened to $n$ tokens such that $F_t \in \mathbb{R}^{n \times c}$. In a single-head attention layer,

$$Q = Q^{\text{sup}}T^Q, \ K = F_t T^K, \ V = F_t T^V \tag{2}$$

where $T^Q, T^K, T^V \in \mathbb{R}^{c \times c}$ are weight matrices. We use $Q^{\text{sup}}$ as the query input and the network feature $F_t$ as the key and value inputs. The attention weights $A$ and attention results $\text{attn}$ is given by,

$$A = \text{Softmax}\left(QK^\top\right) \tag{3}$$

$$\text{attn}\left(Q^{\text{sup}}, F_t, F_t\right) = AV \tag{4}$$

In multi-head attention $\text{MHA}()$, $Q^{\text{sup}}$ and $F_t$ is split to $Q_1^{\text{sup}}, \ldots, Q_M^{\text{sup}}$ and $F_{(t,1)}, \ldots, F_{(t,M)}$, where $M$ is the number of heads and every part is split to dimension $c' = c/M$,

$$\tilde{Q}^{\text{sup}} = \left[\text{attn}_1(Q_1^{\text{sup}}, F_{(t,1)}, F_{(t,1)}); \ldots; \text{attn}_M(Q_M^{\text{sup}}, F_{(t,M)}, F_{(t,M)})\right] \tag{5}$$

$$\text{MHA}\left(Q^{\text{sup}}, F_t, F_t\right) = \text{LayerNorm}\left(Q^{\text{sup}} + \text{Dropout}\left(\tilde{Q}L\right)\right) \tag{6}$$

where $\text{LayerNorm}$ is layer normalization [2], $\text{Dropout}$ is dropout operation [53] and $L \in \mathbb{R}^{c \times c}$ is a projection. Passing the result to a feed-forward network which is effectively a two layer linear projection with $\text{ReLU}$ activation followed also by residual connection, $\text{Dropout}$ and $\text{LayerNorm}$ completes the Transformer decoder layer. Stacking multiple layers gives us the affinity feature $F^{\text{aff}} \in \mathbb{R}^{k^{\text{sup}} \times c}$. Then $F^{\text{aff}}$ is linearly projected to the space of supervised keypoints by weight matrix $P$ and transformed using $\text{Softmax}$ function among self-supervised keypoints into affinity matrix,

$$W = \text{Softmax}\left(F^{\text{aff}}P\right) \tag{7}$$

Each row in $W \in \mathbb{R}^{k^{\text{sup}} \times k^{\text{self}}}$ represents the relation between self-supervised keypoints and corresponding supervised keypoint. Typically we have $k^{\text{sup}} \leq k^{\text{self}}$ for higher flexibility. The final supervised heatmaps is given by,

$$H_t^{\text{sup}} = H_t^{\text{self}}W^\top \tag{8}$$

That is, supervised heatmaps are a weighted sum or selection of the self-supervised heatmaps. This presents supervised loss as,

$$\mathcal{L}^{\text{sup}} = \left\|H_t^{\text{sup}} - H_t^{\text{gt}}\right\|^2 \tag{9}$$

where $H_t^{\text{gt}}$ is the ground truth keypoint heatmap of target image built by placing a 2D gaussian at each joint's location [43, 65].

Our Transformer design explicitly models the relation between supervised and self-supervised tasks. Basic feature sharing model, even with the self-supervised task replaced by a similar pose estimation task, still fails to make sure that two tasks will cooperate instead of competing with each other. Learning an affinity matrix aligns self-supervised keypoints with supervised ones, avoiding the conflicts in multi-task training. During Test-Time Personalization, basic feature sharing model often lacks flexibility and is faced with the risk of overfitting to self-supervised task, due to the decoupling structure of two task heads. Our method, however, enforces the coupling between tasks using an affinity matrix and maintains flexibility as typically there are more self-supervised keypoints than supervised ones. Besides, compared to convolution model, Transformer shows superior ability to capture global context information, which is particularly needed when learning the relation between one supervised keypoint and all self-supervised ones.

Finally, we jointly optimize those two tasks during training. For a training sample, besides the supervised task, we randomly choose another sample belonging to the same person as the target to reconstruct. The final loss is given by

$$\mathcal{L} = \mathcal{L}^{\text{sup}} + \lambda \mathcal{L}^{\text{self}} \tag{10}$$

where $\lambda$ is a weight coefficient for balancing two tasks.

## 3.2 Test-Time Personalization

During inference with a specific person domain, we apply Test-Time Personalization by fine-tuning the model relying solely on the self-supervised task. Given a set of $N^{\text{test}}$ images of the same person $I_1^{\text{test}}, \ldots, I_{N^{\text{test}}}^{\text{test}}$, where $N^{\text{test}} > 1$, we first freeze the supervised Transformer part and update the shared encoder $\phi$ and the self-supervised head $\psi^{\text{self}}$ with the reconstruction loss $\mathcal{L}^{\text{self}}$. Then the updated shared encoder $\phi^*$ and self-supervised head $\psi^{\text{self}*}$ are used along with the supervised head for final prediction. Specifically, during prediction, the updated features and self-supervised head will output improved keypoint heatmaps which leads to better reconstruction. These improved self-supervised heatmaps will go through the Transformer at the same time to generate improved supervised keypoints.

During the personalization process, we propose two settings including the *online* scenario which works in a stream of incoming data and the *offline* scenario which performs personalization on an unordered test image set. We illustrate the details below.

**(i) The online scenario**, which takes input as a sequence and requires real-time inference such as an online camera. In this setting, we can only choose both source $I_s^{\text{test}}$ and target $I_t^{\text{test}}$ with the constraint $s \leq T, t \leq T$ at time $T$ for fine-tuning. Prediction is performed after each updating step.

**(ii) The offline scenario**, which has access to the whole person domain data and has no requirement of real-time inference, assuming we have access to an offline video or a set of unordered images for a person. In this setting, we shuffle the images in the dataset and perform offline fine-tuning, and then we perform prediction at once for all the images.

Compared to *online* scenario, *offline* scenario benefits from more diverse source and target sample pairs and avoids the variance drifts when updating the model. Since our method is designed to personalize pose estimation, the model is initialized with diversely trained weights when switching person identity. In each scenario, different re-initialization strategies can also be applied to avoid overfitting to a local minimum. The various combination of scenarios and reinitializing strategies engifts our method with great flexibility.

It should be noted that our method has *no requirement of consecutive or continuous* frames but only unlabeled images belonging to the same person, which is less costly and beyond the reach of temporal methods such as 3D convolution with multiple frames. Test-Time Personalization can be done at inference without annotations and thus is remarkably suitable for privacy protection: The process can be proceeded locally rather than uploading data of your own for annotating for specialization.

## 4 Experiment

### 4.1 Datasets

Our experiments are performed on three human pose datasets with large varieties to prove the generality and effectiveness of our methods. While the datasets are continuous videos, we emphasize that our approach can be generalized to discontinuous images. In fact, we take the datasets as unordered image collections when performing *offline* Test-Time Personalization. All input images are resized to $128 \times 128$ with the human located in the center.

**Human 3.6M** [24] contains 3.6 million images and provides both 2D and 3D pose annotations. We only use 2D labels. Following the standard protocol [74, 34], we used 5 subjects for training and 2 subjects for testing. We sample the training set every 5 frames and the testing set every 200 frames. We use the Percentage of Correct Key (PCK) as the metric and the threshold we use is the 20% distance of the torso length.

**Penn Action** [71] contains 2,236 video sequences of different people. 13 pose joints are given for each sample in the annotations. We use the standard training/testing split and also use PCK with threshold distance of half distance of torso as the evaluation metric.

**BBC Pose** [8] consists of 20 videos of different sign language interpreter. We use 610,115 labeled frames in the first ten videos for training, and we use 2,000 frames in the remaining ten videos (200 frames per video) with manual annotation for testing. The testing frames are not consecutive. The evaluation method of BBC Pose is the joint accuracy with $d$ pixels of ground truth where $d$ is 6 following [9, 12, 48, 26].

Table 1: Evaluation results on pose estimation. Our proposed method is denoted as *Transformer (keypoint)*. For Human 3.6M and Penn Action datasets, mPCK is employed as the metric while for BBC Pose we use mAcc. The proposed method not only performs better on the validation set but also enjoys more gain in Test-Time Personalization.

| Method | TTP Scenario | Human 3.6M | Penn Action | BBC Pose |
|---|---|---|---|---|
| Baseline | w/o TTP | 85.42 | 85.23 | 88.69 |
| Feat. shared (*rotation*) | w/o TTP | 87.37 (+1.95) | 84.90 (−0.33) | 89.07 (+0.38) |
| | Online | 88.01 (+2.59) | 85.86 (+0.63) | 89.34 (+0.65) |
| | Offline | 88.26 (+2.84) | 85.93 (+0.70) | 88.90 (+0.21) |
| Feat. shared (*keypoint*) | w/o TTP | 87.41 (+1.99) | 85.78 (+0.55) | 89.65 (+0.96) |
| | Online | 89.43 (+4.01) | 87.27 (+2.04) | 91.48 (+2.79) |
| | Offline | 89.05 (+3.63) | 88.12 (+2.89) | 91.65 (+2.96) |
| Transformer (*keypoint*) | w/o TTP | 87.90 (+2.48) | 86.16 (+0.93) | 90.19 (+1.50) |
| | Online | 91.70 (+6.28) | 87.75 (+2.52) | **92.51 (+3.82)** |
| | Offline | **92.05 (+6.63)** | **88.98 (+3.75)** | 92.21 (+3.52) |

## 4.2 Implementation Details

**Network Architecture.** We use ResNet [22] followed by three transposed convolution layers as encoder $\phi$. Every convolution layer has 256 channels, consisting of BatchNorm and ReLU activation and upsampling 2 times to generate the final feature $F$ of size $256 \times 32 \times 32$ and $c = 256$. Considering the diversity of datasets, we use ResNet50 for Penn Action and ResNet18 for both Human 3.6M and BBC Pose. We use one convolution layer as the supervised head $\psi^{\text{sup}}$ and another convolution layer for self-supervised head $\psi^{\text{self}}$, where the supervised head $\psi^{\text{sup}}$ denotes the standard structure used in Appendix **??**. For all three datasets, the output channel for self-supervised keypoints is $k^{\text{self}} = 30$. We adopt a 1-layer Transformer with 4 heads and the hidden layer in feed-forward has 1024 dimensions. The weight of self-supervised loss is set to $\lambda = 1 \times 10^{-3}$ for Penn Action and BBC Pose, $\lambda = 1 \times 10^{-5}$ for Human 3.6M.

**Joint Training.** We apply the same training schedule across methods. For all datasets, we use batch size of 32, Adam [31] optimizer with learning rate 0.001 and decay the learning rate twice during training. We use learning schedule [18k, 24k, 28k], [246k, 328k, 383k] and [90k, 120k, 140k] for BBC Pose, Penn Action, and Human 3.6M respectively. We divide the learning rate by 10 after each stage. The training schedule of BBC Pose is shortened since the data is less diverse.

**Test-Time Personalization (TTP).** During Test-Time Personalization, we use Adam optimizer with fixed learning rate $1 \times 10^{-4}$. The supervised head $\psi^{\text{sup}}$ and Transformer are frozen at this stage. Test-Time Personalization is applied without weight reset unless specified. In offline scenario, even though the model can be updated for arbitrary steps, we adopt the same number of steps as the online scenario for a fair comparison. See Appendix **??** for more details.

**Batching Strategy.** In joint training and TTP offline scenario, both target and source images are randomly chosen and are different within a batch. In TTP online scenario, the target images are always the current frame, which are the same within a batch, whereas the source images are randomly chosen from the previous frames and are different in a batch. In all the scenarios, each target-source pair is performed data augmentation with same rotation angle and scale factor for the two images to make reconstruction easier.

## 4.3 Main Results

To better analyze the proposed method, in Table 1 we compare it with three baselines: (i) **Baseline.** The plain baseline trained with supervised loss only. (ii) **Feat. shared (rotation).** Instead of self-supervised keypoint estimation, we use rotation prediction to compute the self-supervised loss $\mathcal{L}^{\text{self}}$ in Eq. 10 following Sun et al. [58]. Rotation is predicted with a standalone supervised head $\psi^{\text{sup}}$. The two tasks have no direct relation except they share the same feature backbone. Weight coefficient $\lambda$ is set to $1 \times 10^{-4}$ for better performance. (iii) **Feat. shared (keypoint).** We use the self-supervised keypoint estimation task [26] as the self-supervised objective. However, supervised keypoints are still estimated with a standalone supervised head $\psi^{\text{sup}}$ instead of our Transformer design. The two tasks are only connected by sharing the same feature backbone. See Appendix **??** for more details. Finally, our proposed method is denoted as **Transformer (keypoint)**.

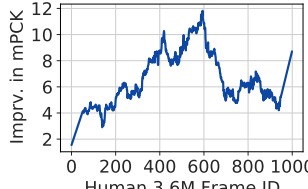 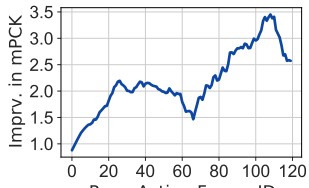 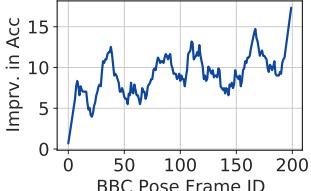

Figure 3: Improvement vs Frame ID in online scenario for 3 datasets. We plot the gap between the Test-Time Personalization and the baseline model for each frame step. We adopt the averaged metric across all test videos. In most cases, we observe TTP improves more over time.

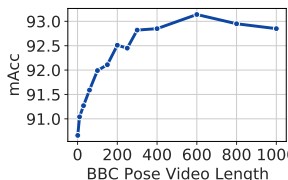 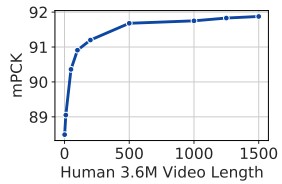

Figure 4: Test-Time Personalization with different numbers of unlabeled test samples. **Left:** mAcc for different video length on BBC Pose. **Right:** mPCK for different video length on Human 3.6M. Test-Time Personalization benefits from utilizing more unlabeled test samples.

Table 2: Test-Time Personalization in online scenario with different update iterations.

| Iters | Penn Action | BBC Pose |
|-------|-------------|----------|
| 1 | 87.75 | 92.51 |
| 2 | **88.01** | **92.64** |
| 3 | 76.27 | 92.59 |
| 4 | 76.13 | 92.53 |

Despite using calibrated self-supervised task weight, *Feat. shared (rotation)* still shows limited and even degraded performance on all three datasets, indicating that a general self-supervised task without a specific design is likely to hurt the performance of supervised one. On the other hand, *Feat. shared (keypoint)* presents superior performance over *Baseline*, even without Test-Time Personalization. This demonstrates the hypothesis that selecting a related or similar self-supervised task can facilitate the original supervised task and thus naturally leads to better performance in Test-Time Personalization. The results of Test-Time Personalization show the personalizing ability of our method. Personalizing for a single person results in significant improvement.

*Transformer (keypoint)* further boosts performance with Test-Time Personalization, with an improvement of 6.63 mPCK on Human 3.6M, 3.75 mPCK on Penn Action, and 3.82 mAcc on BBC Pose. More importantly, our design of learning an affinity matrix not only improves the performance of joint training but also achieves a higher improvement in Test-Time Personalization. For example, TTP in online scenario has an improvement of 2.32 mAcc with *Transformer (keypoint)* compared to an improvement of 1.83 mAcc with *Feat. shared (keypoint)* for BBC Pose. This demonstrates that using the proposed Transformer, two tasks cooperate better in joint training and have higher flexibility in Test-Time Personalization.

In terms of different scenarios for Test-Time Personalization, it is found that the offline scenario does not always surpass online scenario. For example in BBC Pose, both online scenario and offline scenario improve performance, yet in offline scenario, there is a small decrease in mAcc. This is expected for two reasons. Firstly the major advantage of offline scenario comes from the diversity of test samples while BBC Pose has a nonconsecutive validation set selected specifically for diversity. Secondly, we set the learning rate based on the performance of online scenario and follow it in all settings to demonstrates the generality of our method. Better results can be achieved if the learning rate is adjusted more carefully.

## 4.4 Analysis on Test-Time Personalization

**Number of Unlabeled Test Samples.** Our method exploits personal information using unlabeled samples in a single person domain. We observe that more unlabeled samples can further improve the performance in Test-Time Personalization. We study the number of unlabeled samples using extra validation samples of BBC Pose and Human 3.6M. We emphasize that although labels are also provided for extra validation samples, we only use images *without* labels. All experiments have the same setting as Transformer in online scenario and the prediction and evaluation are on the same fixed test set. In Figure 4, we report results of TTP by using different video lengths of samples in fine-tuning in an online manner. For video length smaller than the actual test sequences, we apply

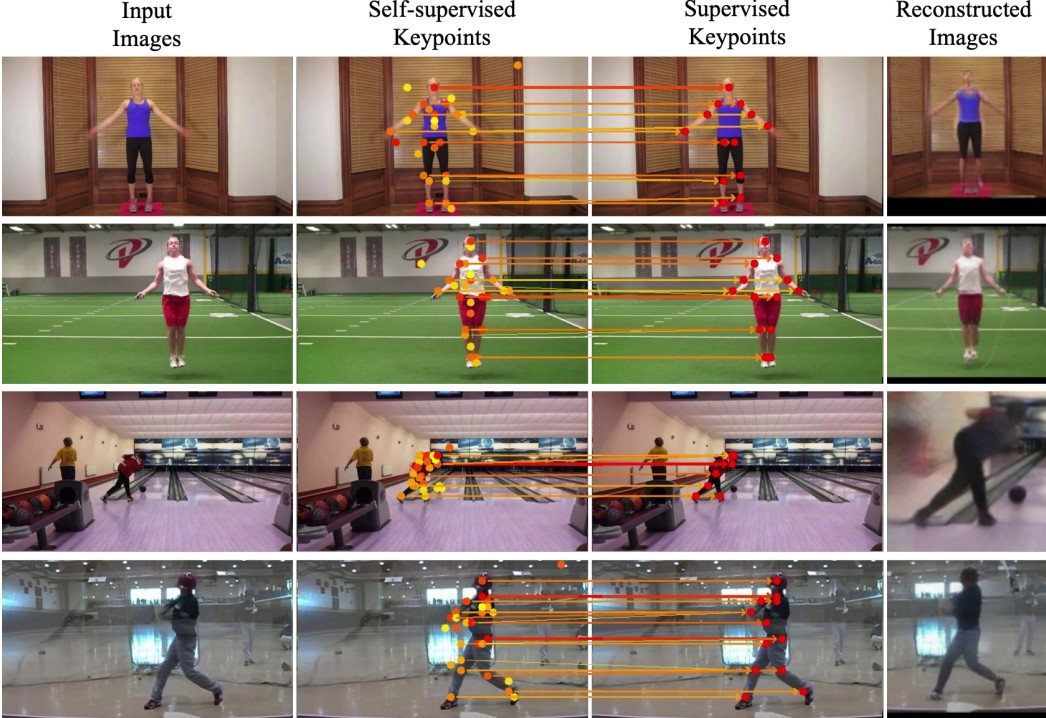

| Input
Images | Self-supervised
Keypoints | Supervised
Keypoints | Reconstructed
Images |

Figure 5: Visualization on Penn Action. The images from the left to the right are: the original image, the image with 30 self-supervised keypoints, the image with 13 supervised keypoints, and the reconstructed image from the self-supervised task. The arrows between keypoints indicate their correspondences obtained from the affinity matrix with the Transformer. Warmer color indicates higher confidence.

reinitializing strategy to simulate shorter videos. We observe that for Human 3.6M, the performance of our model increases as the number of unlabeled samples grows. Similar results appear in BBC Pose except that the performance reduces slightly after using more than 600 frames in fine-tuning. The reason is that the changes of the person images in BBC Pose are very small over time, which leads to overfitting.

**Improvement in Online Scenario.** Figure 3 shows the improvement curve within each test video in the online scenario with respect to the ID ($n$-th update) of frames in TTP. We compute the metric gap between our method using TTP and baseline without TTP for each ID. The results are averaged across all the test videos. In Human 3.6M, we report on a single subject S9. The curves are smoothed to reduce variance for better visualization. The result suggests the gap keeps increasing within a single test video, as the model updates at every frame. Moreover, in later frames, the model has seen more test samples, which helps enlarge the performance gap.

In Human 3.6M, which has much more samples in a single person domain, the performance improves at the beginning but the improvement starts to drop a bit at 600 time steps due to overfitting in later frames. This phenomenon is expected since the examples in Human 3.6M are also quite similar. Note that the gap still exists for later frames, it is only the improvement that becomes smaller.

**Transfer Across Datasets.** We test the generalization of the model by conducting the full training on one dataset and applying TTP on another dataset. We train our model on the Penn Action, and then perform TTP to transfer the model to Human 3.6M. As shown in Table 6, our method using TTP can improve over the baselines by a large margin. Our offline approach achieves around $8\%$ improvement over the baseline without using Transformer and TTP. This shows TTP can significantly help generalize and adapt the pose estimator across data from different distribution.

**Update Iterations.** We show the ablation on update iterations in Table 2. Note that in our online scenario setting, we only update the model once for every incoming test image. We present results where we update more iterations in Table 2. It suggests that more update iterations do not help much. Specifically, for Penn Action the performance drops when we update 3 to 4 iterations. The reason

Table 5: Smoothing results for Penn Action. Our method is complementary to temporal methods.

| Method | TTP Scenario | mPCK | w/ smoothing |
|---|---|---|---|
| Baseline | w/o TTP | 85.23 | 85.68 (+0.45) |
| Transformer | w/o TTP | 86.16 | 86.58 (+0.42) |
| Transformer | Online | 87.75 | 88.31 (+0.56) |
| Transformer | Offline | **88.98** | **89.51 (+0.53)** |

Table 6: The results of transferring the model trained on Penn to Human3.6M.

| Method | TTP Scenario | mPCK |
|---|---|---|
| Baseline | w/o TTP | 52.60 |
| Transformer | w/o TTP | 56.27 |
| Transformer | Online | 60.14 |
| Transformer | Offline | 61.04 |

is, in each step of online setting, we only perform training on one single frame, which can lead to overfitting to a particular image.

## 4.5 Comparisons with Video Models

**Visualization.**  We provide visualization on Penn Action experiments in Figure 5. We visualize the self-supervised keypoints and the supervised keypoints (2nd and 3rd columns). The arrows from the self-supervised keypoints and supervised keypoints indicate the keypoint correspondence, according to the affinity matrix in the Transformer. We show arrows (correspondence) where the probability is larger than 0.1 in the affinity matrix. We use warmer color to indicate larger confidence for both keypoints and arrows. The last column visualizes the reconstructed target image in our self-supervised task, which has the size as the network inputs cropped from the original images. See Appendix **??** for more visualization results.

**Complementary to Temporal Methods.**  Even though our method is designed for single image input and requires no consecutive frames like videos, it is complementary to temporal methods such as 3D convolution [47] or smoothing techniques [48]. We apply Savitzky–Golay filter for smoothing along with our methods for demonstration. In Table 5 we present the results on Penn Action, as Penn Action is the only dataset here with completely consecutive test samples. After smoothing, our method presents a similar improvement to baseline. The increase in accuracy when performing smoothing is too small so the performance gain of our method does not come from temporal information and can be further improved combined with temporal methods.

Table 3: Comparisons with state-of-the-art on Penn Action.

| Method | Penn Action |
|---|---|
| Baseline | 85.2 |
| Ours | 89.0 |
| *video-based methods* | |
| Iqbal et al. [25] | 81.1 |
| Song et al. [52] | 96.5 |
| Luo et al. [39] | 97.7 |

Table 4: Comparisons with state-of-the-art on BBC Pose. Result with (*) is reported in [26]. Ours *(best)* is with extra unlabeled samples.

| Method | BBC Pose |
|---|---|
| *Charles et al. [9] | 79.9 |
| Baseline | 88.7 |
| Ours | 92.5 |
| Ours *(best)* | 93.1 |
| *video-based methods* | |
| Pfister et al. [48] | 88.0 |
| Charles et al. [10] | 95.6 |

**Comparisons with State-of-the-Art.**  In Table 3 and Table 4 we compare our best results with state-of-the-art models on Penn Action and BBC Pose. Note that most of the methods on both datasets use multiple video frames as inputs and use larger input resolutions, which makes them not directly comparable with our method. We report the results for references. We argue that our approach with single image input has more flexibility and can be generalized beyond video data. Most works on Human 3.6M focus on 3D pose estimation thus are not reported.

## 5  Conclusion

In this paper, we propose to personalize human pose estimation with unlabeled test samples during test time. Our proposed Test-Time Personalization approach is firstly trained with diverse data, and then updated during test time using self-supervised keypoints to adapt to a specific subject. To enhance the relation between supervised and self-supervised tasks, we propose a Transformer design that allows supervised pose estimation to be directly improved from fine-tuning self-supervised keypoints. Our proposed method shows significant improvement over baseline on multiple datasets.

**Societal Impact.** The proposed method improves the performance on the human pose estimation task, which has a variety of applications such as falling detection, body language understanding, autonomously sports and dancing teaching. Furthermore, our method utilizes unlabeled personal data and thus can be deployed locally, reducing human effort and avoiding privacy violation. However, the proposed method can also be used for malicious purposes, such as surveillance. To avoid possible harmful applications, we are committed to limit our methods from any potential malicious usage.

**Acknowledgments and Funding Transparency Statement.** This work was supported, in part, by gifts from Qualcomm, TuSimple and Picsart.

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
