# OpenReview forum: "Test-Time Personalization with a Transformer for Human Pose Estimation"
_NeurIPS.cc/2021/Conference — NeurIPS 2021 Poster_

### Official Review · Reviewer_TuGV · 2021-07-12

**Rating:** 6
**Confidence:** 3

**Summary:**

This paper proposes to train personalized human pose estimators for different people by combining ideas from test-time training [57] with self-supervised keypoint estimation [26]. There is a self-supervised component during training using an image reconstruction task based on [26], and additionally, a transformer is trained in a supervised fashion to map the self-supervised keypoints to supervised keypoints (illustrated in Figure 2). There is a decent set of experiments performed on common human pose datasets - Human3.6M, Penn Action, and BBC Pose. Results demonstrate that the proposed method that is tuned for each subject is able to outperform baselines on pose estimation metrics.

**Limitations And Societal Impact:**

I appreciate the paragraph the authors added at the end of the paper on societal impacts, but I’m not clear as to what the authors meant in line 359: “we are committed to limit our methods from any potential malicious usage”. How do you plan on doing this? Seems the method could also be used to train personalized pose detectors without a person’s knowledge or consent (although supervised pose estimation methods have this limitation too, but the proposed method could make this more efficient).

**Main Review:**

Strengths -
1. While elements of the proposed method has been studied in existing works (ex: [57], [26]), applying it to test-time personalization of human pose estimation is a sufficiently new application, and the authors also studied new elements, such as the transformer to map unsupervised keypoint estimates to supervised keypoints.
2. The experiments do demonstrate the authors’ claims that test-time personalization improves pose estimation performance. While it is not surprising that personalizing pose estimators can lead to better results, there is a number of additional experiments in Section 4.4 & 4.5 that provide a few more insights on the model (but I have some additional questions - see weaknesses).

Weaknesses -

1. I am wondering how camera viewpoint changes, occlusion, and actions taken by the subject during the online or offline test-time personalization phase affect the results. In particular, there are some poses in Human3.6M with heavy occlusion, where I anticipate the self-supervised keypoints would not work well. In cases such as this, is the performance of test-time personalization also lower? I feel the paper would be stronger with some comments about this aspect.
2. The model performance seems sensitive to some parameters. For example, Table 2 shows a big drop in performance on Penn Action with more iterations (would be good to show the effect of iterations on Human3.6M). Overfitting also seems to be an issue based on the experiments on Section 4.4.
3. No error bars are reported, and only a single evaluation run seems to be done.


**Time Spent Reviewing:**

3.5

---

> ### Author Response · Authors · 2021-08-11
> **Comment on reviewer TuGV's review**
>
> We thank the reviewer for the thoughtful comments. In the following, we seek to address each of the concerns.
>
> **Q:** *“I am wondering how camera viewpoint changes, occlusion, and actions taken by the subject during the online or offline test-time personalization phase affect the results. In particular, there are some poses in Human3.6M with heavy occlusion [...]”*
>
> **A:** Apart from Human 3.6M, there are actually poses with occlusion in Penn Action as well. In our experiments, we show that our approach is actually robust to and it can help the model adapt to these cases. Please see “**Discussion on Occlusions and Unusual Poses**” in our general comments for more details.
>
> ---
>
> **Q:** *“The model performance seems sensitive to some parameters. For example, Table 2 shows a big drop in performance on Penn Action with more iterations (would be good to show the effect of iterations on Human3.6M). Overfitting also seems to be an issue based on the experiments on Section 4.4.”*
>
> **A:** While the performance drops on Penn Action when training for more iterations, it is a reasonable outcome since the videos in Penn Action are very short. We perform additional experiments with Human 3.6M which has longer videos, and also the offline setting in Penn Action in the following table. We find in both cases the model performance maintains similar with more iterations.
>
> | Iters		| Penn Action (offline) | Human 3.6M (online) |
> | ----------------- | --------------------------- | ---------------------------- |
> | 1		| 88.98			| 91.70			|
> | 2		| 89.24			| 91.78			|
> | 3		| 89.00			| 91.73			|
> | 4		| 89.04			| 91.79			|
>
> ---
>
> **Q:** *“No error bars are reported, and only a single evaluation run seems to be done.”*
>
> **A:** We did not report error bars because the results are stable enough empirically. To show that, we re-ran some experiments of our method on Penn Action in Table 1. The results are as follows. (run1 is the original run reported in the paper)
>
> | TTP Scenario | run1	 	| run2	 	| run3	 	| variance	|
> | ----------------- | ----------------- | ----------------- | ----------------- | ----------------- |
> | w/o TTP	| 86.16		| 86.28		| 86.18		| 0.0028	|
> | online TTP	| 87.75		| 87.64		| 87.74		| 0.0025	|
>
> ---
>
> **Q:** *“I appreciate the paragraph the authors added at the end of the paper on societal impacts, but I’m not clear as to what the authors meant in line 359: “we are committed to limit our methods from any potential malicious usage”. How do you plan on doing this?”*
>
> **A:** We thank the reviewer for bringing up this concern. As we discuss in L. 214 - 216, in application, one can personalize the pose estimator completely offline on their edge device, without uploading the images to the server, thus preserving privacy. If our method is used in products, the personal information must be acquired with the person’s knowledge and consent, as in other personalization applications such as music recommendation systems.

---

> > ### Comment · Reviewer_TuGV · 2021-08-24
> > **Thanks for your response**
> >
> > I appreciate the authors for addressing most of my concerns and the additional experiments. I have read the author responses and the other reviews, and I will keep my current rating (borderline, lean towards accept). This is because while the authors addressed most of my concerns, I feel the other reviewers bring up a valid point regarding background variations. I understand that the dataset do contain some background variations, but it is not as much as real/in-the-wild data. Adding a dataset such as MPI-INF-3DHP which includes outdoor scenes could make the paper stronger.

---

### Official Review · Reviewer_Ng6V · 2021-07-17

**Rating:** 6
**Confidence:** 4

**Summary:**

This paper proposes a 2D human pose estimation learning framework that uses self-supervision for test-time personalization. The model is first trained with labeled data with both a self-supervised and a supervised task, and then is fine-tuned only with the self-supervised task at test time with unlabeled personalized data. The paper shows consistent improvement over the fully supervised baseline on three datasets under three scenarios (without, with online, and with offline test-time personalization).

**Ethical Concerns:**

N/A.

**Limitations And Societal Impact:**

Yes.

**Main Review:**

My main concern about the paper is the novelty (as a NeurIPS submission). The test-time personalization framework, and each component used in this framework (image reconstruction with pose and appearance features, Transformer-based attention masking, etc.) have been proposed in previous literature, the paper focuses on their application to the human pose estimation problem.

The paper claims that test-time personalization allows models to generalize better to out-of-distribution data, which is the key point of doing test-time personalization. However, I think the experiments do not sufficiently back up this claim. One missing experiment to show this can be to conduct full training (self-supervised and supervised) on one dataset (e.g., Human3.6M), and apply test-time personalization on a different dataset (e.g., Penn Action). In this case, it would also be interesting to compare with a baseline that trains with labeled data on the main domain (e.g., Human3.6M) with/without a small amount of labeled data on the target domain (e.g., Penn Action).

Line 126, the paper mentions by limiting the heatmap channel number to avoid encoding appearance features. However, this does not guarantee removing appearance features in F_t, which is fed to the Transformer. This potential appearance feature leakage can potentially cause test-time personalization to fail when the supervised domain and the unsupervised test domain have large discrepancy. In this sense, the experiment mentioned in my second point is necessary.

Line 142, any citation for the claim “similar-looking tasks do not necessarily help each other even when sharing features”? Also, it is not clear to me how this leads to the proposal for using a Transformer decoder to “further strengthen their coupling”. Please clarify.

The writing of the test-time personalization online scenario (line 199-201) is not very clear to me. When we have a new testing image at time T, do we run multiple training steps on all the data before time T, or do we only need to train one or a few steps only with this testing image at T? I assume it is the latter (thus the “Update Iteration” experiment in Table 2), because otherwise I wonder how real-time prediction can be possibly achieved. Then it is confusing to me that the paper also mentions using data from s <=T and t <= T (line 201). Please clarify this in Section 3.2.

The statement that “the gap keeps increasing” on line 311 but “starts to drop” on line 315 is confusing. Looking at Figure 3, the gaps always increase and drop for the three datasets.

What architecture is used for the appearance encoder \phi^{app}?

What happens when there is only one image for a target person at test time?

The current method focuses on single-person pose estimation, which I assume relies on person detection. Any thoughts on how to apply this to multi-person problems (detection, pose estimation, etc.)?


**Time Spent Reviewing:**

3

---

> ### Author Response · Authors · 2021-08-11
> **Comment on reviewer Ng6V's review**
>
> We thank the reviewer for the thoughtful comments. In the following, we seek to address each of the concerns.
>
> **Q:** *“My main concern about the paper is the novelty (as a NeurIPS submission). The test-time personalization framework, and each component used in this framework (image reconstruction with pose and appearance features, Transformer-based attention masking, etc.) have been proposed in previous literature.”*
>
> **A:** We respectfully disagree with the reviewer. One could always say a component of a framework has existed before, as most deep learning papers use ConvNets but it does not mean they do not have novelty. While the test-time training framework already exists, one fundamental problem as stated in LN32-33 is: One task does not necessarily help another by just sharing features. This is a common problem that exists in multi-task learning as well.
>
> Our key contribution and innovation is: Instead of hoping two **parallel** tasks can help each other, we re-organize one task (supervised keypoint) to depend on another task (unsupervised keypoint) in a **sequential** manner, using a Transformer. To our knowledge, while Transformer has a lot of applications, it has not been used to condition or transfer one task upon another. This exact design allows fine-tuning one task to improve another, which we consider suits well for test-time training and novel.
>
> ---
>
> **Q:** *“One missing experiment to show this can be to conduct full training (self-supervised and supervised) on one dataset (e.g., Human3.6M), and apply test-time personalization on a different dataset (e.g., Penn Action).”*
>
> **A:** We thank the reviewer for the great suggestions! Indeed, we find TTP has shown much larger advantage when personalization is performed across datasets. We train our model on the Penn Action dataset, and then perform TTP to transfer the model to Human 3.6M dataset, using both online and offline settings. As shown in the following table, our method using TTP significantly improves over baselines.
>
> | Method	| TTP Scenario	| Penn Action -> Human 3.6M |
> | ----------------- | --------------------------- | ----------------- |
> | Baseline	| w/o TTP		| 52.60		|
> | Ours		| w/o TTP		| 56.27		|
> | Ours		| online			| 60.14		|
> | Ours		| offline			| 61.04		|
>
> ---
>
> **Q:** *“Line 126, the paper mentions by limiting the heatmap channel number to avoid encoding appearance features. However, this does not guarantee removing appearance features in F_t, which is fed to the Transformer.”*
>
> **A:** We first clarify that besides limiting the channel number, a more fundamental reason that pose structural bottleneck is created is because the model use spatial softmax to regularize the output to be keypoints. While $F_t$ can still contain appearance information, it is completely fine. For keypoints, the appearance is removed by softmax. For the Transformer, it provides a way to transfer one task to another task. As long as the output of the Transformer is translating the self-supervised keypoints to supervised keypoints (both do not contain appearance), it does not matter what information the Transformer takes as inputs.
>
> ---
>
> **Q:** *“Line 142, any citation for the claim “similar-looking tasks do not necessarily help each other even when sharing features”? Also, it is not clear to me how this leads to the proposal for using a Transformer decoder to “further strengthen their coupling”. Please clarify.”*
>
> **A:** It is a common phenomenon in multi-task learning that sharing features alone does not mean two similar tasks can help each other. For example [1] shows two pixel prediction tasks can hurt each other with naive feature sharing. As we discussed in the first question, instead of hoping two **parallel** tasks can help each other, we re-organize one task (supervised keypoint) to depend on another task (unsupervised keypoint) in a **sequential** manner, using a Transformer. In this way, the changes of self-supervised keypoints will directly be transformed to the supervised keypoint, thus “further strengthen their coupling”.
>
> [1] Misra et al, Cross-stitch networks for multi-task learning, CVPR 2016.
>
> ---
>
> **Q:** *“The writing of the test-time personalization online scenario (line 199-201) is not very clear to me. When we have a new testing image at time T, do we run multiple training steps on all the data before time T, or do we only need to train one or a few steps only with this testing image at T? I assume it is the latter (thus the “Update Iteration” experiment in Table 2), because otherwise I wonder how real-time prediction can be possibly achieved. Then it is confusing to me that the paper also mentions using data from s <=T and t <= T (line 201). Please clarify this in Section 3.2.”*
>
> **A:** Your assumption is partially correct. We do update for one step at time T. However, note that updating at time T does not restrict to only using the one new testing image at time T. In real-world deployment, we can stash the past images (data from s <=T and t <= T) to buffer, and update the model at each timestep by sampling images from the buffer.
>
> ---
>
> **Q:** *“The statement that “the gap keeps increasing” on line 311 but “starts to drop” on line 315 is confusing. Looking at Figure 3, the gaps always increase and drop for the three datasets.”*
>
> **A:** We believe the gap only drops in the end for Human 3.6M (left). For BBC Pose (right), the gap appears to be increasing consistently. For Penn Action (middle), the gap begins at 1.0. It increases to around 2.0 in 20 - 80 frames and finally rises to about 3.0. It only drops very slightly after about 110 frames. Note that the videos are different in lengths. The number of available videos becomes smaller for later timesteps, hence larger variance.
>
> ---
>
> **Q:** *“What architecture is used for the appearance encoder \phi^{app}?”*
>
> **A:** $\phi^{app}$ is a simple CNN composed of 8 convolutional layers. We have also uploaded our code for reference.
>
> ---
>
> **Q:** *“What happens when there is only one image for a target person at test time?”*
>
> **A:** When there is only one image, we can use data augmentation to obtain different copies of the same image for TTP.
>
> ---
>
> **Q:** *“The current method focuses on single-person pose estimation, which I assume relies on person detection. Any thoughts on how to apply this to multi-person problems (detection, pose estimation, etc.)?”*
>
> **A:** State-of-the-art approaches for multi-person pose estimation like Mask R-CNN also require human detection. Specifically, it first detects the human instance, and then performs single-person pose estimation. Our method can be easily integrated into the second stage of the Mask R-CNN framework when applied to multi-person problems.

---

> > ### Comment · Reviewer_Ng6V · 2021-08-30
> > **Thanks for the responses**
> >
> > I would like to thank the authors for clarifying the paper and adding experiments as suggested. The additional results from training and testing on different datasets are interesting and do clear out concerns about the model adapting to appearance variations to some degree. I have increased my rating to **"6: Marginally above the acceptance threshold"**. In the meantime, I still have some reservations and suggestion about the paper.
> >
> > 1. The paper shows convincing results that the proposed personalization process improves from  the baseline. However, the proposed method cannot directly take advantage of the existing large-scale supervised static datasets (e.g., COCO), as raised by reviewer RVKM. In this sense, an experiment that shows comparison between the proposed personalization and an out-of-the-box state-of-the-art pose estimator trained on COCO would be necessary. Because if the latter is better, then the practicality of the proposed method will be diminished.
> >
> > 2. Following this idea, I wonder if the proposed method can be at least pretrained on static data. It seems to me that it is possible to pretrain the supervised task and the upper keypoint branch of the self-supervised task on COCO (using only L^{sup}). If this is possible, then experiments with such pretraining can potentially make the results more compelling.
> >
> > 3. For the experiment for training/testing on different datasets, I think an even more convincing way to conduct is to swap the current train and test set, and use Human3.6M for training, and Penn Action for testing, as Penn Action contains more background variation than Human3.6M.

---

### Official Review · Reviewer_mHXQ · 2021-07-18

**Rating:** 7
**Confidence:** 4

**Summary:**

Paper proposes to personalize a human pose estimator during test time without any supervision. This is achieved through training model with self-supervised and supervised keypoint detection objectives jointly. During test time, the model is finetuned using the self-supervised objective with test instances. Doing so ensures personalized keypoint estimations. Results on video 2D pose datasets supports that personalization is helpful.

**Limitations And Societal Impact:**

Limitations are not discussed properly. It is advised in the main review to provide elaborate discussion of limitations and failures.

**Main Review:**

## Strengths

- Overall, I find the idea neat and refreshing. It is a good application of test time optimization for human pose with a novel technical contribution i.e. usage of transformer for affinity betweeen self-supervised keypoints and semantic keypoints.

- Paper is clear, easy to follow, and nicely written. SupMat provides and explanatory video and the code.

## Weaknesses

### Experiments:

- I appreciate the current set of experiments and analysis conducted on 3 different datasets. However, I think that the selection of datasets is a bit poor. L.21 motivates that such a test time personalization (TTP) is can be helpful for occlusion and unusual poses. But none of these datasets are challenging enough to prove this point. H36M and PennAction have shots where almost full body of the subjects are visible in all frames. BBC only has upper body annotations. PoseTrack dataset as an example is much more rich in terms of challenging scenarios. Results on PoseTrack can improve the presentation of the idea.

- I am a bit confused about the results presented in the supmat pdf and video. Do they show the results before or after TTP? Can you provide before and after results to be able to see the improvement with TTP. Can you also provide results on videos with occlusion and more challenging poses to evaluate the motivation in L.21?

- Authors should provide failure examples and discuss the limitations of the proposed method.

### Other comments:

- L.125: Is this operation differentiable? could you elobarate what this replacement operation look like?

- As a suggestion, the real potential of this method can be shown by training on a large scale unlabeled video dataset with only L_self. Then, learned self-supervised keypoints can be finetuned to a specific dataset through an additional training with L_self + L_sup. This can potentially yield more robustness and improved performance.

- L.320. Did you perform the same experiment with offline TTP?

- EFT[1] is a method that proposes test time optimization for 3D human body regression. Even though the idea is fundamentally different, authors can still discuss it in the related work section.

[1] Joo et al, Exemplar Fine-Tuning for 3D Human Pose Fitting Towards In-the-Wild 3D Human Pose Estimation, Arxiv 2020.

### Post Rebuttal
I appreciate the authors' effort to reply reviewer comments. Several concerns i.e. weak baseline, background variation, limited novelty are raised by reviewers. Authors responded them in a satisfactory way by providing experimental evidence when needed. Even though the experimental setting can be improved, I see this paper as a proof of concept in the direction of personalizing pose estimation methods. Hence, I decided to increase my score to "7: good paper, accept".

**Time Spent Reviewing:**

6 hours

---

> ### Author Response · Authors · 2021-08-11
> **Comment on reviewer mHXQ's review**
>
> We thank the reviewer for the thoughtful comments. In the following, we seek to address each of the concerns.
>
> **Q:** *“[...] However, I think that the selection of datasets is a bit poor. [...] Results on PoseTrack can improve the presentation of the idea.”*
>
> **A:** We thank the reviewer for the suggestion. We emphasize that the datasets we used actually contain videos with changing backgrounds and are challenging. Please check “**Discussion on Background Variation**” and “**Discussion on Occlusions and Unusual Poses**” in our general comments. We agree that PoseTrack provides challenging scenarios. However, PoseTrack is a multi-person pose estimation dataset. Our TTP approach, as a first step using self-supervision to personalize human pose estimation, primarily focuses on single human video. How to deal with multiple person videos will be an interesting direction in future work.
>
> ---
>
> **Q:** *“I am a bit confused about the results presented in the supmat pdf and video. Do they show the results before or after TTP? Can you provide before and after results to be able to see the improvement with TTP? Can you also provide results on videos with occlusion and more challenging poses to evaluate the motivation in L.21?”*
>
> **A:** The visualizations are before TTP. We show the before and after results in https://drive.google.com/file/d/1u-XEkwZaKBXMQFPfFr6JefrZWcE0P5Pi/view?usp=sharing . By fine-tuning the reconstruction, TTP improves the model’s recognition on human joints. For instance, in the first example, the keypoint estimation of the shoulder of the human is corrected after using TTP. For examples with occlusions and unusual poses, please see: https://drive.google.com/file/d/1EYumkHOH8LGWQ6605qhe7vFWcOCELzlm/view?usp=sharing and “**Discussion on Occlusions and Unusual Poses**” in our general comments.
>
> ---
>
> **Q:** *“Authors should provide failure examples and discuss the limitations of the proposed method.”*
>
> **A:** For failure examples, see the following link: https://drive.google.com/file/d/1eLsuGKOhRGj5a3QQoYDxWwI-rhhw75M0/view?usp=sharing
>
> In failure example (A), the model confuses between left and right foot. However, this confusion still leads to reasonable reconstruction with the self-supervised task since the two legs have similar appearance. Thus the reconstruction loss here does not help correct the prediction result. For failure example (B), similarly, the arm and the background have similar appearance, thus when the keypoint is localized to background, the reconstruction still remains reasonable.
>
> ---
>
> **Q:** *“L.125: Is this operation differentiable? could you elobarate what this replacement operation look like?”*
>
> **A:** It is differentiable. In L. 124 we have computed the condensed keypoints from $H_t^{self}$ using softmax, i.e. the 2-d location of each keypoint. Suppose we have $k$-th keypoint located at $(x, y)$, then the $k$-th channel of $F_t^{kp}$ is computed by $exp(- \frac{(x-i)^2 + (y-j)^2}{2 \sigma^2})$, where $(i, j)$ is the pixel location. The overall operation is differentiable because both softmax and the gaussian pdf are differentiable. The operation guarantees that only the 2-d location information is passed through the information bottleneck so that the model cannot cheat.
>
> ---
>
> **Q:** *“As a suggestion, the real potential of this method can be shown by training on a large scale unlabeled video dataset with only L_self. Then, learned self-supervised keypoints can be finetuned to a specific dataset through an additional training with L_self + L_sup.”*
>
> **A:** We thank the reviewer for the suggestion. We agree that it is likely to yield more robustness and improved performance, and it is easy to extend from our current method. While our current work focuses on personalization during test-time, a better pre-training should be complementary to our method and it will be an interesting future direction.
>
> ---
>
> **Q:** *“L.320. Did you perform the same experiment with offline TTP?”*
>
> **A:** We provide additional results for the same experiments with offline TTP on Penn Action datasets in the following table. We observe that TTP is robust to different number of iterations in offline experiments.
>
> | Iters		| Penn Action (offline) 	|
> | ----------------- | ----------------- |
> | 1x		| 88.98		|
> | 2x		| 89.24		|
> | 3x		| 89.00		|
> | 4x		| 89.04		|
>
> ---
>
> **Q:** *“EFT[1] is a method that proposes test time optimization for 3D human body regression. Even though the idea is fundamentally different, authors can still discuss it in the related work section.”*
>
> **A:** Thank you for pointing out the paper. We will add it to our related work in the revised version.
>
> [1] Joo et al, Exemplar Fine-Tuning for 3D Human Pose Fitting Towards In-the-Wild 3D Human Pose Estimation, Arxiv 2020.

---

> > ### Comment · Reviewer_mHXQ · 2021-08-28
> > **Thanks for the responses**
> >
> > I appreciate the authors' effort to reply reviewer comments. Several concerns i.e. weak baseline, background variation, limited novelty are raised by reviewers. Authors responded them in a satisfactory way by providing experimental evidence when needed. Even though the experimental setting can be improved, I see this paper as a proof of concept in the direction of personalizing pose estimation methods. Hence, I decided to increase my score to "7: good paper, accept".

---

### Official Review · Reviewer_RVKM · 2021-07-22

**Rating:** 5
**Confidence:** 5

**Summary:**

A model is proposed to perform 2d human pose estimation from a monocular image. To train, the model requires 2 images from the same video of the same person: one supervised with a 2d pose and one without supervision. The system simultaneously learns self-supervised keypoints and supervised keypoints, as well as a transformation between them using a transformer network. During inference, this design allows fine-tuning of some parts of the network with additional non-annotated images from the video. The results show that this self-supervised personalization scheme brings improvements over the only-supervised baseline.

**Limitations And Societal Impact:**

Please see Weaknesses in Main Review.


**Main Review:**

Strengths:
1) Personalizing human pose estimation is an important topic.
2) The methodology is well explained, self-contained, which makes it easy to follow.
3) The way the splitting of the self-supervised and supervised keypoint estimation is performed (using a transformer to explicitly model their relation using an affinity matrix) is a nice idea.

Weaknesses:

My main concern is the practicality of this approach:

1) First, it has no training regime for learning from datasets of only supervised static images (for example, COCO). This means that it cannot benefit from the large diversity offered by these datasets.
2) Second, the method is also limited to datasets where the source and target images have small background variation. This is the case during both training and inference personalization. The following argument is being made: “our method has no requirement of consecutive or continuous frames but only unlabeled images belonging to the same person”. Yet, there is no experiment where the same person comes from different videos with different backgrounds (or even just camera motion).
3) Third, there is no discussion on speed. How does video personalization compare to video-based methods which are far more accurate? Would TTP be practical?
4) Fourth, the results are far from the state of the art. While I understand that the setup is not the same, this is caused mostly by the limitations of the method (e.g.  necessity to use small input resolution).

Clarity:
1) It should be clear from the abstract that 2d (not 3d) pose estimation is of interest here.
2) L242:  $\Psi^{sup}$ is not defined at this point.
3) Batching Strategy (from Supp Material) should go into the main paper. It is an important detail without which it is difficult to understand how TTP works.
4) How is $H_t^{gt}$ built?
5) L227-228: [1] actually uses half the distance of the head as a threshold in PCKh@0.5. Is this what you are calculating and reporting?
6) Complementary to Temporal Methods : The increase in accuracy when performing smoothing is too small so say that “the performance gain of our method does not come from temporal information”. It is also contradicting this statement “After smoothing, our method presents a similar improvement to baseline”. Can you please clarify?


Post Rebuttal:

I thank the other reviewers for their comments and the authors for their answers! I have decided to maintain my rating: 5: Marginally below the acceptance threshold.
This is mainly motivated by the current method not being able to use annotated static images during training and not convincing enough experiments for background variation.


**Time Spent Reviewing:**

12

---

> ### Author Response · Authors · 2021-08-11
> **Comment on reviewer RVKM's review**
>
> We thank the reviewer for the thoughtful comments. In the following, we seek to address each of the concerns.
>
> **Q:** *“First, it has no training regime for learning from datasets of only supervised static images (for example, COCO). This means that it cannot benefit from the large diversity offered by these datasets.”*
>
> **A:** The datasets where we conducted experiments (BBC Pose and Human 3.6M) do contain videos showing the same subject with large background variation, as discussed in our general comment on **Discussion on Background Variation**. This indicates that our method can be applied on static images with same subject, different backgrounds. However, there is no such dataset publicly available, thus we use diverse video datasets. Collecting such a dataset will be an interesting future direction.
>
> ---
>
> **Q:** *“Second, the method is also limited to datasets where the source and target images have small background variation. [...] Yet, there is no experiment where the same person comes from different videos with different backgrounds (or even just camera motion).”*
>
> **A:** As we discussed in the general comment on **Discussion on Background Variation**, while the examples we showed in the supplementary pdf are mostly Penn Action video frames with similar backgrounds, there are actually obvious background changes in the BBC pose and we can find camera changes in the Human 3.6M dataset.
>
> ---
>
>
> **Q:** *“Third, there is no discussion on speed. How does video personalization compare to video-based methods which are far more accurate? Would TTP be practical?”*
>
> **A:** We emphasize our method is not constrained by the backbone model, and it is a complementary technique to video-based backbone. We have proposed both the offline and online scenario in our paper, and they are both practical: (i) For the offline scenario, given a collection of images for a particular person, we first finetune the model using TTP on these images before the actual inference. Since this is a pre-processing step, it is practical to spend some amount of time. (ii) For the online scenario, the speed of finetuning and inference on each frame together is about 4.15 FPS on a single RTX 3090 GPU. However, this can be easily accelerated by adopting a lower finetuning rate. For example, in the following table, we show online TTP results on Penn Action with 8 times lower update rate (we update the model once every 8 incoming images). This leads to roughly real-time inference. The performance gain over baseline remains significant.
>
> | Method   | TTP Scenario | TTP rate | Penn Action |
> |----------|--------------|----------|-------------|
> | Baseline | w/o TTP      |          | 85.23       |
> | Ours     | online       | 1        | 87.75       |
> | Ours     | online       | 1/8      | 87.23      |
>
> ---
>
> **Q:** *“Fourth, the results are far from the state of the art. While I understand that the setup is not the same, this is caused mostly by the limitations of the method (e.g. necessity to use small input resolution).”*
>
> **A:** Our setup is not the cause of lower performance, and our method is not limited to lower resolution input. We conduct experiments using lower resolution due to limited computing resources, given large amounts of ablation studies. We report the results of our method using higher 256x256 resolution in the following table, which is on par with the state of the arts. We emphasize that our goal is not to beat the best number but rather show TTP can achieve a large gain in a general framework.
>
> | Method	| TTP Scenario	| BBC Pose	| Human3.6M |
> | ----------------- | --------------------------- | ----------------- | ----------------- |
> | Baseline	| w/o TTP		| 92.34		| 92.25            |
> | Ours		| w/o TTP		| 93.71		| 92.23            |
> | Ours             	| online	                        | 95.82             | 93.72            |
> | Ours              | offline                        | 97.18             | 95.00            |
>
> ---
>
> **Q:** *“It should be clear from the abstract that 2d (not 3d) pose estimation is of interest here.”*
>
> **A:** Thank you for your advice. We will add it in the revised version.
>
> ---
>
> **Q:** *“L242: $\psi^{sup}$ is not defined at this point.”*
>
> **A:** We are sorry for the confusion. $\psi^{sup}$ actually denotes a structure used in one of the baselines denoted as “Feat. shared (keypoint)” in Table 1. $\psi^{sup}$ is further illustrated in Figure 1 of the supplementary pdf. We will explain it carefully in the revised version.
>
> ---
>
> **Q:** *“Batching Strategy (from Supp Material) should go into the main paper. It is an important detail without which it is difficult to understand how TTP works.”*
>
> **A:** Thank you for the advice. We will add it to Sec. 4.2 in the revised version.
>
> ---
>
> **Q:** *“How is $H_t^{gt}$ built?”*
>
> **A:** Following [42,64], $H_t^{gt}$ is built by placing a 2D gaussian at each joint’s location.
>
> [42] Newell et al, Stacked hourglass networks for human pose estimation, ECCV 2016.
>
> [64] Xiao et al, Simple baselines for human pose estimation and tracking, ECCV 2018.
>
> ---
>
> **Q:** *“L227-228: [1] actually uses half the distance of the head as a threshold in PCKh@0.5. Is this what you are calculating and reporting?”*
>
> **A:** We are using PCK@0.2 for Human 3.6M and Penn Action. We are sorry for the typo and appreciate you for pointing it out. We will revise Sec. 4.1 in a later version.
>
> ---
>
> **Q:** *“The increase in accuracy when performing smoothing is too small so say that “the performance gain of our method does not come from temporal information”.*
>
> **A:** Thanks for the comment. We agree with this sentiment and will rephrase the sentence.
>
> ---
>
> **Q:** *“It is also contradicting this statement “After smoothing, our method presents a similar improvement to baseline”. Can you please clarify?”*
>
> **A:** Thanks for the comment. The purpose of Table 3 is to show our method is complementary to temporal smoothing. What we meant is that we can obtain consistent improvement over baseline using TTP no matter whether we perform temporal smoothing or not. We will clarify in our paper.

---

> > ### Comment · Reviewer_RVKM · 2021-08-30
> > **Thank you for the responses!**
> >
> > I appreciate the new experiments with larger resolution and sampling fewer frames in the online setting. The new results on the chosen datasets are good.
> >
> > Still, my concern about accommodating for datasets containing static images with very high pose, background, occlusion variations (MPII Human Pose, COCO) is not answered. Being limited to training pose estimation only in this setup (by requiring at least 2 images of the same person) can be problematic, as a lot of annotation work already performed by the community cannot be used. It is not clear to me that a system estimating challenging human poses can do away with this limitation.
> >
> > I therefore maintain my rating: 5: Marginally below the acceptance threshold.

---

> > > ### Author Response · Authors · 2021-08-30
> > > **Thank you for the replies!**
> > >
> > > We answer the question on the COCO dataset in our first reply above. Our answer is more focusing on how our method is applied to diverse images with high background, occlusion variations, but not on the single image aspect. We would like to discuss more here.
> > >
> > > Since our work is focusing on “personalization”, this application generally assumes we have a collection of images from a particular person, and we can personalize the model to this person. Please also see previous work from [11] on similar applications.
> > >
> > > Nevertheless, our model can be easily generalized and applied to a single-image testing case. We can use data augmentation to obtain different copies of the same image, and then feed them into our pipeline for Test-Time Personalization. While there is not enough time to perform this experiment towards the end of the discussion. We will add this experiment in our final version of the paper.

---

### Author Response · Authors · 2021-08-11
**General Comment**

We thank all the reviewers for your comprehensive and insightful comments. We will revise our paper according to your suggestions. In the following, we will first address some common concerns. We will then address each of your concerns in separate comments.

**Discussion on Background Variation**

Our method works quite well with large variation backgrounds. We show in the following link that there are actually videos with different backgrounds in both the Human 3.6M and BBC Pose dataset (backgrounds change between source and target images): https://drive.google.com/file/d/1gVzmbEdl37wT3SfkdSRYU3Wrb0Opgbs8/view?usp=sharing . In Human 3.6M, images of each subject can come from **different cameras**. In BBC Pose,  the background plays news videos, which are changing frequently. From these examples, we observe that most of the self-supervised keypoints are focusing on the foreground human. We conjecture that the model tends to localize the self-supervised keypoints on foreground to achieve better reconstruction on the places that can be modeled.

---

**Discussion on Occlusions and Unusual Poses**

We find our method TTP indeed improves results on cases with occlusions and unusual poses. We show examples with partial occlusions and unusual poses in https://drive.google.com/file/d/1EYumkHOH8LGWQ6605qhe7vFWcOCELzlm/view?usp=sharing . In both examples in Human 3.6M and Penn Action, we can see the pose estimation (supervised) results got improved after using TTP, especially on the human legs.

Quantitatively, we evaluate on Penn Action on how much TTP can improve over baselines on the visible keypoints when the others are occluded. In the following table, we show that for images with less than 8 visible keypoints, which are 15.8% of all the validation images, our method has a performance gain over the baseline of 4.76 on the visible keypoints. The performance gain over the rest of the images is  3.68 over the baseline. Specifically, for images with only 6 keypoints visible, our accuracy on these keypionts is 5.56 better than baseline. These all show our method has a better advantage when the test instance is presented with occlusion.

| Number of visible joints | Imprv. | Percentage in valid set |
|--------------------------|--------|------------|
| 6                        | 5.56   | 3.1        |
| <8                       | 4.76   | 15.8       |
| >=8                      | 3.68   | 84.2       |

---

### Decision · Program_Chairs · 2021-09-27

**Decision:**

Accept (Poster)

**Comment:**

This work addresses the problem of test time personalization for human keypoint estimation from images. It builds on existing self-supervised keypoint discovery methods (e.g. [26]) by adding a Transformer-based adaptor for learning the transformation between self-supervised and supervised keypoints. The novelty lies in the use of a learned affinity matrix that maps between the two representations. During personalization, the model is adapted to the test set in either an online or offline setting. Quantitative results are presented on three different human pose datasets.

There were concerns in the reviews regarding the choice of datasets used. For instance, in Human3.6M, while there can be changes induced by the different cameras, the overall variation is very small and is highly correlated with the individual cameras. One reviewer was also concerned that the method, as presented, could not be trained on the large numbers of existing static datasets (e.g. COCO) due to the need for source and target frames featuring the same individual. The choice of datasets directly influence the difficulty of the task e.g. the range of poses, the appearance variation in the backgrounds, etc.

During the discussion, the authors have provided additional experiments investigating occlusion, more training iterations, and cross dataset transfer. These experiments were valuable for better understanding the strengths and weaknesses of the proposed method. In the end, some of the reviewers increased their scores and the majority leaned towards acceptance. The authors are strongly encouraged to include these additional results and explanations in the revised paper along with adding additional discussion of the limitations mentioned by the reviewers of the experiments (e.g. the challenges associated with large changes in background appearance).